# Cephalopod species identification using integrated analysis of machine learning and deep learning approaches

Hui Yuan Tan[1], Zhi Yun Goh[1], Kar-Hoe Loh[2], Amy Yee-Hui Then[3], Hasmahzaiti Omar[3,4] and Siow-Wee Chang[1]

[1] Bioinformatics Programme, Institute of Biological Sciences, Faculty of Science, Universiti Malaya, Kuala Lumpur, Malaysia

[2] Institute of Ocean & Earth Sciences, Universiti Malaya, Kuala Lumpur, Malaysia

[3] Ecology Biodiversity Programme, Institute of Biological Sciences, Faculty of Science, Universiti Malaya, Kuala Lumpur, Malaysia

[4] Museum of Zoology, Institute of Biological Sciences, Faculty of Science, Universiti Malaya, Kuala Lumpur, Malaysia

## ABSTRACT

**Background**. Despite the high commercial fisheries value and ecological importance as prey item for higher marine predators, very limited taxonomic work has been done on cephalopods in Malaysia. Due to the soft-bodied nature of cephalopods, the identification of cephalopod species based on the beak hard parts can be more reliable and useful than conventional body morphology. Since the traditional method for species classification was time-consuming, this study aimed to develop an automated identification model that can identify cephalopod species based on beak images.

**Methods**. A total of 174 samples of seven cephalopod species were collected from the west coast of Peninsular Malaysia. Both upper and lower beaks were extracted from the samples and the left lateral views of upper and lower beak images were acquired. Three types of traditional morphometric features were extracted namely grey histogram of oriented gradient (HOG), colour HOG, and morphological shape descriptor (MSD). In addition, deep features were extracted by using three pre-trained convolutional neural networks (CNN) models which are VGG19, InceptionV3, and Resnet50. Eight machine learning approaches were used in the classification step and compared for model performance.

**Results**. The results showed that the Artificial Neural Network (ANN) model achieved the best testing accuracy of 91.14%, using the deep features extracted from the VGG19 model from lower beak images. The results indicated that the deep features were more accurate than the traditional features in highlighting morphometric differences from the beak images of cephalopod species. In addition, the use of lower beaks of cephalopod species provided better results compared to the upper beaks, suggesting that the lower beaks possess more significant morphological differences between the studied cephalopod species. Future works should include more cephalopod species and sample size to enhance the identification accuracy and comprehensiveness of the developed model.

Corresponding authors
Kar-Hoe Loh, khloh@um.edu.my
Siow-Wee Chang, siowwee@um.edu.my

## INTRODUCTION

Cephalopods (Phylum: Mollusca, Class: Cephalopoda) refer to a group of soft-bodied, bilaterally symmetrical animals with well-developed head and body, which includes octopus, squid, cuttlefish and nautilus. This taxon is the third largest molluscan class which comprises more than 800 described species in the world (*Lindgren et al., 2004*). Globally, cephalopods contribute as much as 55% of fishery landings and 70% in fishery value (USD) (*Hunsicker et al., 2010*). Their economic contribution as fisheries resources has been on the rise globally as the landings of finfish had decreased due to overfishing. Cephalopods also play an important role in marine food webs, particularly in supporting top marine predators such as sharks and dolphins (*Wolff, 1984*; *Hunsicker et al., 2010*).

In Malaysia, cephalopods contribute to about 12% of total fisheries landing valued at MYR1,067 million (approximately USD 250 million) in retail markets (*Department of Fisheries Malaysia, 2015*). The west coast of Peninsular Malaysia contributed more than 50% of the country's cephalopod landings, and the taxonomic composition was dominated by squids, followed by cuttlefish and octopus (*Abu & Isa, 1986*). Limited local biodiversity surveys found 17 species from six families and common families included Sepiidae (cuttlefish), Octopodidae (octopus), and Loliginidae (squid) (*Samsudin, 2001*; *Rubaie et al., 2012*). Cephalopods are popular in local cuisines such as grilled squids (known locally as 'sotong bakar'), fried calamari, and dried squids. Despite their high fisheries profile and economic value as well as ecological importance, there has been virtually no comprehensive taxonomic studies on cephalopods in Malaysia.

One major reason for the lack of local taxonomic information on cephalopods is due to their soft body mass which renders morphological descriptions based on length measurements very challenging. The soft tissue nature makes these animals easily damaged during sampling and rapidly digested inside the stomachs of predators; only the intact upper and lower chitinized beaks will typically remain as evidence of their consumption (*Markaida & Hochberg, 2005*). It had been established that the morphological characteristics of these beaks are species-specific and thus allow for taxonomic identification of the cephalopod prey from beaks isolated from stomach contents of their predators (*Clarke, 1962*; *Clarke, 1986*; *Furness, Laugksch & Duffy, 1984*). The lower beak displays greater inter-specific morphological variations than the upper beak; thus the former is generally used for identification purpose (*Xavier, Phillips & Cherel, 2011*). The inclusion of beak size information has been shown useful to differentiate between two cephalopod species from the same family (*Clarke, 1962*).

Machine learning approaches such as Artificial Neural Network (ANN), k-Nearest Neighbors (kNN), Random Forest (RF), and Support Vector Machine (SVM) are used increasingly to automate taxonomic identification efforts and to improve accuracy of classification tasks. Many taxonomic studies to date focused on the application of machine learning in the identification of plant species (*Tan et al., 2020*; *Murat et al., 2017*), land animals (*Nguyen et al., 2017*; *Norouzzadeh et al., 2018*), and insects (*Thenmozhi, Dakshayani & Srinivasulu, 2020*) while a small number of marine-related studies had been conducted with a focus on fish identification. Examples of these studies included the use of

machine learning and deep learning methods for tracking and estimation of fish abundance (*Marini et al., 2018*), identification of fish species using whole-body images (*Allken et al., 2018*) and using otolith contours in fish species identification (*Salimi, SK & Chong, 2016*).

Very limited machine learning techniques had been applied on cephalopod classification problems. *Orenstain et al. (2016)* used whole-body laboratory images of cuttlefish for classification of camouflaging behaviour with SVM classifiers. Beak and statolith images were used in the identification of three squid species found in the South China Sea (*Jin et al., 2017*). *Himabindu, Jyothi & Mamatha (2017)* identified 50 squid species based on morphometric features measured from whole-body samples using ANN classifier. These examples showed the usefulness of machine learning methods for identification of cephalopods but their application had been limited to a single taxon (either cuttlefish or squid).

Hence, in this study, all three taxa of cephalopods namely squid, cuttlefish, and octopus were included. Images of whole body, upper beak and lower beak for all samples were taken. These images were then pre-processed using feature descriptors that extract useful information and omit extraneous ones. Specifically, traditional morphometric features were extracted from the images using three feature extraction methods, *i.e.,* grey histogram of oriented gradient (HOG), colour HOG and morphological shape descriptors (MSD). For comparison, deep features were extracted using three convolutional neural networks (CNN) models namely VGG19, InceptionV3 and Resnet50. Next, a cephalopod species classification tool was developed using an integrated analysis of morphometric, machine learning and deep learning approaches. Eight machine learning algorithms were analysed and benchmarked which included ANN, SVM, RF, kNN, Decision Tree (DT), Logistic Regression (LR), Linear Discriminant Analysis (LDA), and Gaussian Naïve Bayes (GNB). The proposed automated identification model will reduce time needed for future cephalopod identification work and increase identification accuracy with reduced costs. Our work also provides baseline estimate of species richness of cephalopods in Peninsular Malaysia which is important for documentation of the country's rich marine biodiversity.

## MATERIALS & METHODS

Generally, there were five main steps involved in this study namely sample collection, image acquisition, image processing, machine learning identification and model evaluation. Figure 1 shows the proposed framework of the cephalopod species identification model using an integrated analysis of machine learning and deep learning approaches. For sample collection, seven cephalopod species were acquired from fisheries landing sites and brought back to the laboratory for subsequent processing including beak extraction. For image acquisition, the images of whole body, upper beaks and lower beaks were captured by using a smartphone. In the image processing step, methods such as pre-processing, rescaling, and segmentation were performed on the images acquired, followed by feature extraction. In the machine learning identification step, eight machine learning methods were used to classify the cephalopod species. Finally, in the model evaluation step, both confusion matrix and precision–recall curve were used to evaluate the performance of the eight machine learning models.

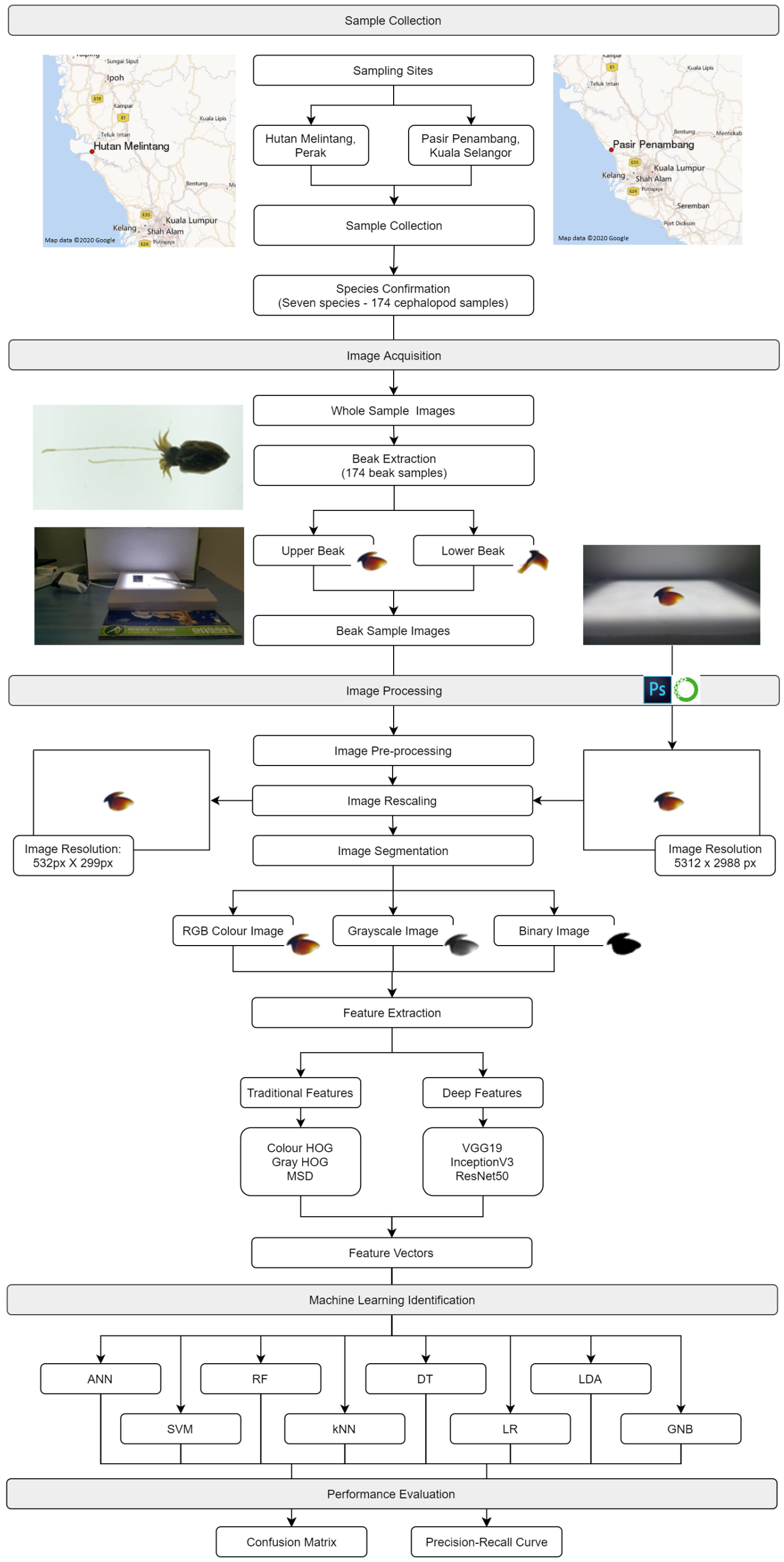

**Figure 1** The framework for cephalopod species identification using integrated analysis of machine learning and deep learning.

## Sample collection

Sampling trips were conducted from November 2017 to January 2018. A total of 174 cephalopod samples were collected from two major fisheries landing sites located on the west coast of Peninsular Malaysia, namely Hutan Melintang, Perak and Pasir Penambang, Kuala Selangor. The specimens were selected to represent the diversity of distinctive morphological groups and size classes available during sampling. Seven putative cephalopod species were selected including four putative species for squid ($n = 96$), two for cuttlefish ($n = 49$) and one for octopus ($n = 29$). Samples collected were kept on ice in the field and frozen immediately upon arrival in the laboratory.

## Species identification and confirmation

After defrosting, the specimens were measured for dorsal mantle length (mm) and wet body mass (g) and were photographed (see steps described below). Initial species identification using morphological characters were conducted using available taxonomic references (*Reid, Jereb & Roper, 2005*; *Jereb et al., 2016*) and were cross-checked against current species checklists in reputable databases, such as the Malaysia Biodiversity Information System (*Malaysia Biodiversity Information System, 2020*), the World Register of Marine Species (*MolluscaBase, 2020*), and SeaLifeBase (*Palomares & Pauly. Editors, 2020*). The upper and lower beaks were then extracted from the buccal mass and preserved separately in labelled jars containing 80% alcohol.

Species identification from morphological characteristics was subsequently confirmed with molecular approaches using the mitochondrial 16S rRNA gene which was amplified with the universal primers 16Sar and 16Sbr (*Simon et al., 1994*).

## Software and hardware

The hardware used for the image acquisition included a lightbox, a laptop (Intel i7 with 4 GB RAM), and a smartphone (16MP camera with 1440×250 pixels). The Adobe Photoshop CS6 (version 13.1.2) and Python 3 software were used in the feature extraction and identification step.

## Image acquisition

The images of both upper and lower beaks were captured using the Samsung Galaxy Note 4 smartphone camera. Samples were placed against a white background, *i.e.*, lightbox with white light, and were centred on the camera screen to ensure high-quality images. A distance of 10 cm between the smartphone camera and the beak samples was fixed and no zooming was used for all specimens photographed (Fig. 2). Only the left lateral views of the upper and lower beaks were captured. All beak images were stored using the JPEG format.

## Image processing and feature extraction

A digital image has to be processed before performing feature extraction and computational analysis to enhance the image, remove the noises, minimize the amount of storage needed and to increase the computation performance. Firstly, unwanted parts in the images were cropped. Next, all the beak images were downscaled to 10% of its original size to eliminate unnecessary pixel information without compromising the quality of the images. The

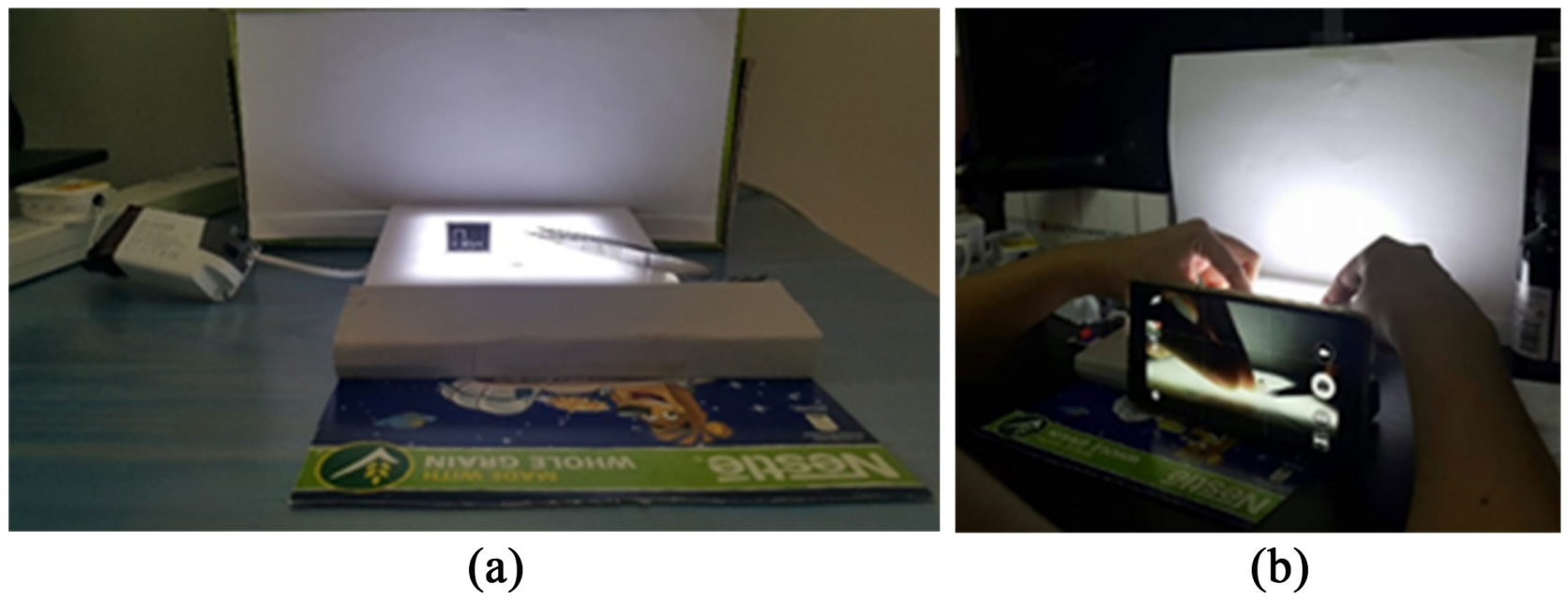

**Figure 2** **Set up for the image acquisition for beak samples of the studied cephalopods: (A) Lightbox; (B) the smartphone was used to capture the photos of the beaks.**

original image resolution was 5312px ×2988px while the rescaled image resolution was approximately 532px ×299px. Gaussian smoothing was then carried out on the rescaled images to reduce the noises on the images.

Image segmentation was performed to convert the rescaled images into red-green-blue (RGB) colour, grayscale and binary images to obtain the region of interest (ROI), which is the beak. Firstly, the colour space of the image was converted to the common RGB colour space. The colour image was then converted to a grayscale image by using Eq. (1) (*OpenCV, 2020*).

$$Y = 0.299(R) + 0.587(G) + 0.114(B). \quad (1)$$

A thresholding algorithm was used to convert the grayscale image into a binary image by classifying all the pixels into two groups using a threshold value set at 240. Those below the threshold value (0–239) formed the image background and those above the threshold (240–255) formed the object of interest, *i.e.,* the beak. The boundary of the beak (ROI) was then determined, outlined and extracted from the image.

Next, HOG feature descriptor was calculated by determining the occurrences of gradients and orientation in localized portions of each image, *i.e.,* an image is broken down into small regions and the gradients and orientation were calculated for each region. The magnitude of the gradient increases wherever there is a sharp change in intensity (brightness) of the image. The magnitude of the gradient, G was computed using Eq. (2) (*Mallick, 2016*).

$$|G| = \sqrt{G_x^2 + G_y^2}. \quad (2)$$

The gradient orientation, was computed using Eq. (3) where $G_x$ is horizontal gradient in the X direction and $G_y$ is vertical gradient in the Y direction (*Mallick, 2016*).

$$\Theta = \arctan\frac{G_x}{G_y}. \quad (3)$$

A single image was divided into several cells, where each cell formed a square region which contained 128×128 pixels each. Using calculated gradient magnitude and orientation,

every pixel within a cell casted a weighted vote for an orientation based histogram. The orientation bin was set as nine and this generated a frequency table for nine angles, *i.e.*, 0, 20, 40, 60, 80, 100, 120, 140 and 160 (*Singh, 2019*). Using the set number of bins, a 9×1 matrix histogram was counted for each cell. Next, 2×2 cells were grouped to form a block, resulting in a total of 3 (3×1) blocks of 128×128 pixels for each image. Each block contained four 9×1 or a single 36×1 matrix and a total of 108 (3× 36×1) features were produced for each image. In this manner, HOG was used in extracting the colour features from grayscale and RGB images and converted them into feature datasets.

The MSD consisted of both geometrical and morphological features (*Aakif & Faisal Khan, 2015*) and extracted the shape features from the highlighted region in the binary images. The binary image is used commonly due to clear delineation of the boundary of the object. Firstly, the image was computed to determine the contour of the object. Then, all the noises were eliminated by adjusting the threshold. After the contour was shown, ten morphological features, including beak size, were extracted from each image (Table 1).

In addition to the traditional methods, three deep learning CNN models, namely VGG19, InceptionV3 and ResNet50, were used to automatically extract features from the images. The VGG19 model consists of 3×3 convolutional layers. The original beak images were firstly resized to 224×224 pixels of RGB images as the inputs for VGG19. The VGG19 model pre-processed the images by subtracting the mean RGB value from each pixel of the image. Next, the max-pooling layer was used to reduce the feature dimensionality and two fully connected layers were used to produce the feature vector, with the presence of 4,096 neurons (*Mateen et al., 2019*).

The InceptionV3 model requires the input RGB image size of 299×299 pixels. Unlike the fixed kernel size for VGG19, the InceptionV3 allowed the features of the images to vary within an image frame and include multiple sizes of kernels in the same layer. There are four operations constructed in parallel, including a 1×1 convolutional layer for depth reduction, a 3×3 convolutional layer for capturing distributed features, a 5×5 convolutional layer for capturing the global features and a max-pooling layer for capturing the low-level features. Each layer is responsible to extract the deep features from the images, concatenate and pass them to the next layer (*Anwar, 2019*).

The ResNet50 model requires the input RGB image size of 224×224 pixels. ResNet50 is composed of 48 convolutional layers with 7×7 and 1×1 kernel size, max pooling layer and the fully connected layer. One of the important features of ResNet50 is the shortcut connections, that skip one or more layers, in order to solve the problem of vanishing gradient in deep neural networks by allowing the gradient to flow through the layer (*Wen, Li & Gao, 2020*).

## Machine learning identification

Machine learning techniques are capable of analysing weighty amounts of image data accurately and successfully. The supervised machine learning methods used for the cephalopod classification problem were Artificial Neural Network (ANN), Support Vector Machine (SVM), Random Forest (RF), Decision Tree (DT), k-Nearest Neighbours (kNN),

**Table 1** List of morphological features.

| Features | Definition | Formula |
|---|---|---|
| Area | Size of the beak | – |
| Perimeter | The length of the contour of the beak | – |
| Aspect Ratio | The ratio of major axis length over minor axis length | $\frac{width\ (w)}{height\ (h)}$ |
| Extent | The proportion of pixels in the bounding box that also contains the beak. | $\frac{area}{bounding\ box\ area}$ |
| Solidity / convexity | The proportion of the pixels in the convex hull that also contains beak. | $\frac{area}{convex\ hull\ area}$ |
| Equivalent Diameter | The diameter of a circle with the same area as the beak | $\frac{(4 \times area)}{pi}$ |
| Circularity | The ratio of the area of the beak to the convex circle | $\frac{(4 \times pi \times area)}{(convex\ perimeter)^2}$ |
| Rectangularity | The ratio of the beak to the area of the minimum bounding rectangle | $\frac{w \times h}{bounding\ box\ area}$ |
| Form Factor | The ratio of the area of the beak to the circle | $\frac{(4 \times pi \times area)}{perimeter^2}$ |
| Narrow Factor | The ratio of the diameter of the beak to the height of the beak | $\frac{equivalent\ diameter}{h}$ |

**Table 2** List of parameters adjusted for each classifier.

| Classifiers | Parameters |
|---|---|
| ANN | Hidden layer sizes = 1 layer and 30 hidden neurons, Learning rate schedule for weight updates = 0.001, Maximum number of iteration = 200, Weight optimization = stochastic gradient-based optimizer |
| SVM | $C$ = 30, Decision function shape = One-vs-one decision function, Kernel type = 'sigmoid' |
| RF | Number of trees = 100, Function in measuring the quality of split = 'Gini impurity', Maximum number of features = sqrt(number of features) |
| DT | Function in measuring the quality of split = Information gain, Maximum depth of the tree = 2 |
| kNN | Distance metric = minkowski (standard Euclidean metric), Number of neighbors = 8, Weights function = uniform weights |
| LR | $C$ = 0.15, Multiclass = multinomial loss fit across the probability distribution, Weight optimization algorithm = newton-cg |
| LDA | $C$ = 0.15, Weight optimization = Singular value decomposition, tol = 0.0001 |
| GNB | Largest variance for calculation stability = 1e−09 |

**Notes.**

C, regularization parameter based on the squared l2 penalty, the smaller the strong regularization power; tol, tolerance for stopping criteria.

Logistic Regression (LR), Linear Discriminant Analysis (LDA), and Gaussian Naïve Bayes (GNB). Table 2 shows the list of parameters adjusted for each classifier.

The ANN method is a machine learning technique that processes information by adopting the way neurons of human brains work and consists of a set of nodes that imitates

the neuron and carries activation signals of different strengths (*Daliakopoulos, Coulibaly & K. Tsanis, 2005*).

The SVM method is another popular algorithm in solving classification problems with limited amount of data. First, each sample data is plotted as a point in $n$-dimensional spaces, where $n$ is the number of features obtained from the image, also known as the support vectors. The SVM algorithm finds the best-fit hyperplane that maximizes the margin between the nearest support vectors of both classes with the hyperplane chosen (*Yu & Kim, 2012*).

A decision tree (DT) model is constructed for each feature selected. It starts from the root node and splits at the leaf nodes. Each leaf node determines the best split approach while the Gini impurity function is used to measure the quality of the split. The final leaf node shows the final prediction of each DT (*Fan, Ong & Koh, 2006*).

The RF method is a meta-estimator that fits several decision trees (DTs). The RF model is constructed and trained by the bagging method (decision tree) while the result is based on majority voting. All the predictions resulting from each DT are voted and the majority is the final output of the RF model (*Svetnik et al., 2003*).

For the kNN model, each data point is coordinated in the $n$-dimensional space while an unknown sample is introduced, the distance between the unknown sample with each data point is calculated based on the Euclidean distance matrix (*Alimjan et al., 2018*).

The multinomial LR model has two or more discrete outcomes where the most frequent species in the dataset was chosen as the reference category while the probability of other categories was compared against the reference category. This resulted in $n$–$1$ binary regression models, where $n$ = number of species in the classification problem. Prediction of the LR model is based on the Maximum Likelihood Estimation (MLE) approach, where the MLE determines the mean and variance that best describe the sample (*Brownlee, 2019*).

The LDA algorithm assumes that each feature has the same variance and calculates the between-class and within-class variance (*Balakrishnama & Ganapathiraju, 1998*). The LDA approach maximizes the between-class variance and minimizes the within-class variance. Once the LDA model is trained, the probability of an unknown sample is calculated by using the Bayes Theorem (*Hamsici & Martinez, 2008*). The result of the prediction is chosen based on the highest probability of the species.

The normal distribution method is also applied in the GNB model that estimates the mean and standard deviation of each species from the training data given. The Bayes Theorem is applied to calculate the probabilities of each species. Species with the highest probability matched will be selected (*Pattekari & Parveen, 2012*).

The features extracted from each feature extraction methods were used as the input datasets in the machine learning identification step. Each dataset was split into 80% of training and 20% of the testing set. The training set was used to trained all eight models by minimizing the error formed while the model performance was evaluated using the testing dataset. Five-fold cross-validation (CV) stratified shuffle split method was used and tested for 10 times to avoid any overfitting. A significant feature of stratified shuffle split is the ability to preserve the ratio of the training and testing set in the dataset. Figure 3 shows an example of sample splitting for one of the ANN models.

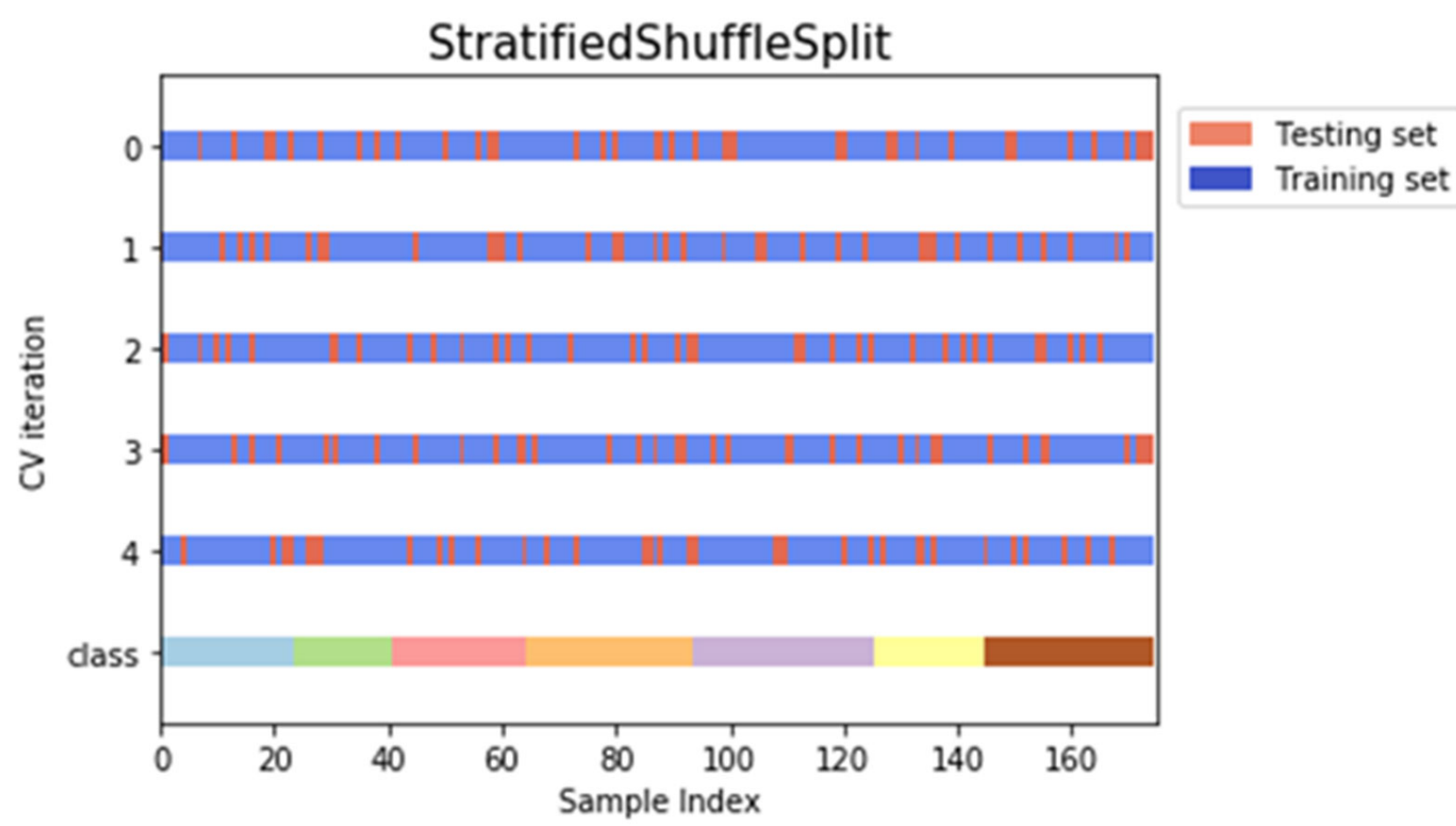

**Figure 3** **Example of stratified shuffle split cross-validation approach for one of the ANN models.**

### Performance evaluation

The performance of the classification model was evaluated through the confusion matrix, *i.e.,*a table that displays the number of true positives (TP), true negatives (TN), false positives (FP) and false negatives (FN). The testing accuracy, precision and recall were calculated using Eqs. (7) to (9) respectively.

$$\text{Testing accuracy} = \frac{\text{TP} + \text{TN}}{\text{TP} + \text{TN} + \text{FP} + \text{FN}} \tag{7}$$

$$\text{Precision} = \frac{TP}{TP + FP} \tag{8}$$

$$\text{Recall} = \frac{TP}{TP + FN} \tag{9}$$

In addition, the performance of each classifier model was visualized from the area under the Precision-Recall (PR) curve (AUC), where the precision value was plotted on the $y$-axis while the recall value at the $x$-axis (*Narkhede, 2018*). The precision and recall values were computed from the testing set for each cephalopod species and the average AUC of each model was calculated. The higher the AUC, the better the model performance in identifying cephalopod species from the beak images. According to *Saito & Rehmsmeier (2015)*, the PR curve is more informative and reliable than the Receiver Operating Characteristic (ROC) curve for imbalanced datasets.

## RESULTS

### Data Collection and Feature Extraction

Seven species of cephalopod were morphologically identified (Table 3) and confirmed through 16S rRNA sequencing (Table 4). Out of these, two species were cuttlefish (*Sepia*

**Table 3** List of the Cephalopod Species Collected.

| Scientific Name (Common Name) | Upper Beak Image* | Lower beak Image* | Sample size |
|---|---|---|---|
| 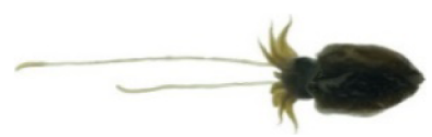 *Sepia aculeata* (Needle cuttlefish) | 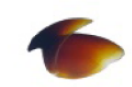 Size ≈ 1cm | 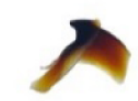 Size ≈ 1cm | 25 |
| 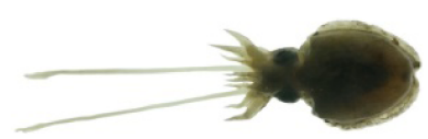 *Sepia esculenta* (Golden cuttlefish) | 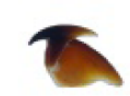 Size ≈ 1cm | 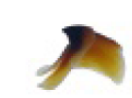 Size ≈ 1cm | 24 |
| 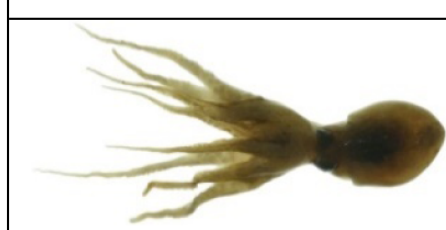 *Amphioctopus aegina* (Sandbird octopus) | 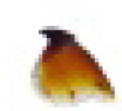 Size ≈ 0.5cm | 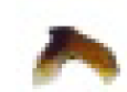 Size ≈ 0.5cm | 29 |
| 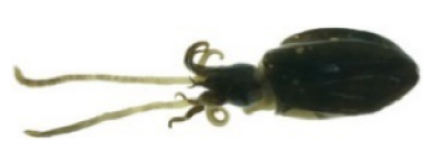 *Sepioteuthis lessoniana* (Bigfin reef squid) | 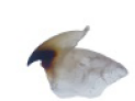 Size ≈ 1cm | 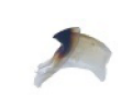 Size ≈ 1cm | 17 |
| 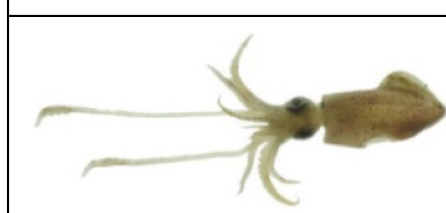 *Loliolus uyii* (Little squid) | 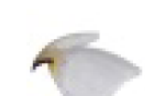 Size ≈ 0.5cm | 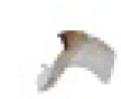 Size ≈ 0.5cm | 32 |
| 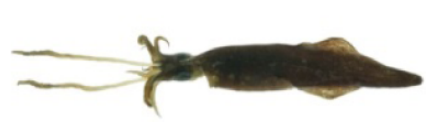 *Uroteuthis chinensis* (Mitre squid) | 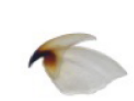 Size ≈ 0.5cm | 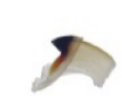 Size ≈ 0.5cm | 19 |
| 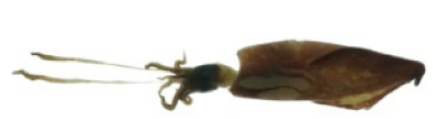 *Uroteuthis edulis* (Swordtip squid) | 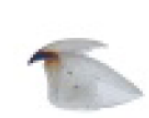 Size ≈ 0.5cm | 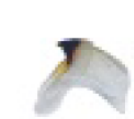 Size ≈ 0.5cm | 28 |

**Notes.**

*Images are not to scale.

*aculeata*, and *Sepia esculenta*), four were squids (*Sepioteuthis lessoniana, Loliolus uyii, Uroteuthis chinensis* and *Uroteuthis edulis*) and one was an octopus (*Amphioctopus aegina*).

A range of 10 to 108 features such as shape and colour features were extracted from the left lateral beak images using HOG and MSD descriptors. Using deep learning methods of VGG19, ResNet50 and InceptionV3, 2048 to 4096 features were extracted. Table 5 lists the number of traditional and deep features extracted by each descriptor.

**Table 4** Seven cephalopod species with GenBank accession number.

| Species | Sample code | Sequence ID | GenBank accession number |
|---|---|---|---|
| *Sepia aculeata* | C2-1 | SeqC2-1 | MZ413930 |
| *Sepia esculenta* | C6-25 | SeqC6-25 | MZ413931 |
| *Sepioteuthis lessoniana* | C3-1 | SeqC3-1 | MZ413932 |
| *Loliolus uyii* | S1-1 | SeqS1-1 | MZ413933 |
| *Uroteuthis chinensis* | S3-1 | SeqS3-1 | MZ413934 |
| *Uroteuthis edulis* | S4-1 | SeqS4-1 | MZ413935 |
| *Amphioctopus aegina* | O2-6 | SeqO2-6 | MZ413936 |

**Table 5** Number of traditional features and deep features extracted.

| Descriptors | Number of features |
|---|---|
| Gray HOG | 108 |
| Colour HOG | 108 |
| MSD | 10 |
| Gray HOG + MSD | 118 |
| Colour HOG + MSD | 118 |
| VGG19 | 4096 |
| ResNet50 | 2048 |
| InceptionV3 | 2048 |

## Traditional features

Firstly, the extracted features were tested individually as a single descriptor of grey HOG, colour HOG and MSD. Each of the descriptors was fit into eight different classifiers to test for the model performance in identifying the cephalopod species. Five-fold cross-validation with 80%–20% stratified shuffle splitting was used to avoid overfitting. Each test was continuously run for 10 times to achieve more reliable results. The testing results were averaged and the PR curves were plotted based on the descriptors used. Table 6 shows the average testing accuracy of each classifier with traditional descriptors for the upper and lower beak images. Grey HOG descriptors achieved the highest testing accuracy of 55.43% (± 0.14) for the kNN model using upper beak images while HOG descriptors with ANN achieved the best accuracy of 68.69% (± 0.12) with lower beak images. MSD descriptors with GNB achieved the best accuracy at 64.00% (± 0.11) using the upper beak images.

Next, the descriptors were combined as the hybrid descriptors which are (grey HOG + MSD) and (colour HOG + MSD) and tested. Table 6 shows the average testing accuracy of each classifier with hybrid descriptors. The hybrid descriptors of (Grey HOG + MSD) and (colour HOG + MSD) with ANN had the best testing accuracy at 61.09% (± 0.14) for the upper beak image and 73.09% (± 0.12) for lower beak images. An improvement in the testing accuracy was observed in comparison with the single descriptor with the increment from 68.69% to 73.09% for the ANN model with lower beak images. Figure 4A shows the confusion matrix from one of the runs of the ANN model for hybrid descriptor (colour HOG + MSD) and lower beak image. The ANN model appeared to classify the test samples

**Table 6** Performance for single and hybrid descriptor of traditional features.

| Model | Testing Accuracy (%)* (AUC) | | | | | | | | | |
|---|---|---|---|---|---|---|---|---|---|---|
| | Gray HOG | | Colour HOG | | MSD | | Gray HOG + MSD | | Colour HOG + MSD | |
| | UB | LB | UB | LB | UB | LB | UB | LB | UB | LB |
| **ANN** | 54.34(0.59) | 52.69(0.59) | 60.40(0.62) | **68.69 (0.76)** | 55.43(0.56) | 44.91(0.51) | **61.09(0.65)** | 58.06(0.65) | 64.91(0.70) | **73.09(0.79)** |
| **SVM** | 46.51(0.56) | 45.31(0.56) | 51.66(0.59) | 66.69(0.74) | 62.17(0.66) | 52.06(0.58) | 52.97(0.61) | 51.14(0.61) | 59.34(0.62) | 68.23(0.77) |
| **RF** | 57.77(0.64) | 51.20(0.57) | 66.06(0.74) | 71.03(0.76) | 64.57(0.69) | 63.66(0.68) | 77.03(0.84) | 65.03(0.72) | 79.14(087) | 76.91(0.81) |
| **kNN** | **55.43 (0.61)** | 48.74(0.55) | 53.77(0.62) | 71.14(0.75) | 61.20(0.66) | 50.63(0.58) | 59.31(0.67) | 50.86(0.58) | 62.69(0.69) | 70.40(0.74) |
| **DT** | 42.40(0.55) | 36.86(0.49) | 45.77(0.61) | 40.29(0.53) | 45.71(0.61) | 46.46(0.63) | 43.31(0.57) | 48.69(0.65) | 44.97(0.62) | 53.14(0.67) |
| **LR** | 50.91(0.58) | 50.63(0.58) | 55.94(0.61) | 65.26(0.75) | 42.00(0.55) | 41.03(0.51) | 60.06(0.66) | 56.63(0.64) | 62.34(0.69) | 69.89(0.78) |
| **LDA** | 57.14(0.58) | 45.83(0.41) | 64.80(0.67) | 61.89(0.61) | 62.74(0.66) | 59.26(0.60) | 74.34(0.74) | 58.23(0.54) | 76.74(0.80) | 64.34(0.62) |
| **GNB** | 41.71(0.51) | 44.97(0.48) | 48.17(0.57) | 53.54(0.57) | **64.00(0.66)** | 52.74(0.60) | 51.66(0.58) | 50.06(0.53) | 54.97(0.62) | 57.49(0.61) |

**Notes.**

*Average testing accuracy from the five-fold CV results with 10 times runs.

UB, Upper beak; LB, Lower beak; AUC, Average area under the precision–recall curve in one of the runs.

Bolded text indicated the best results for each traditional feature model (RF and LDA models showed overfitting as testing accuracy was much lower than the training accuracy).

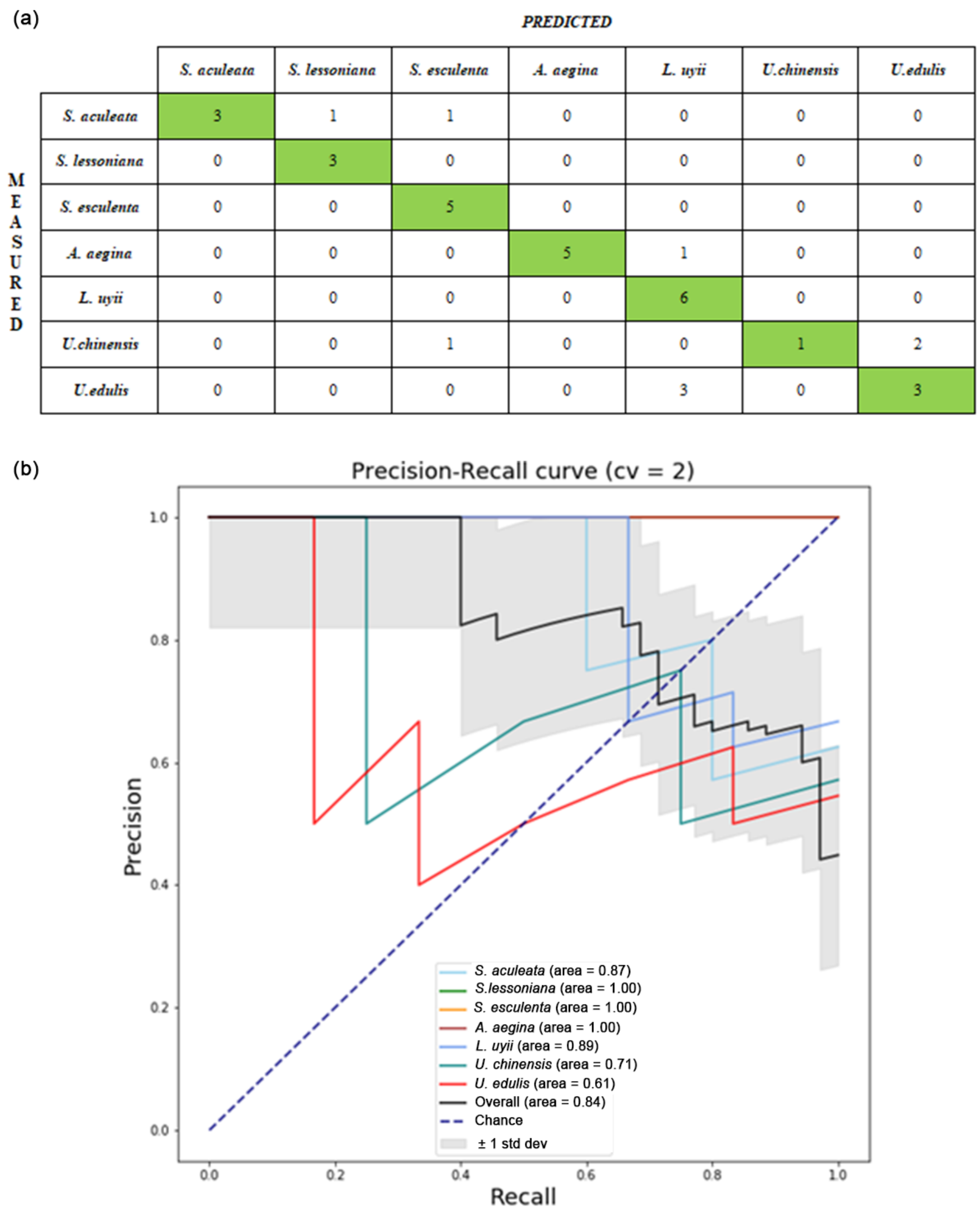

**Figure 4 Performance evaluation from one of the runs in the ANN model with hybrid descriptor (colour HOG +MSD) of lower beak images: (A) confusion matrix; (B) precision-recall curve.** For the confusion matrix, the precision and recall value of the identification model was computed from the testing set. Each cephalopod species was computed for its precision and recall values to visualize the differences in the performance of the model. The average precision–recall curve of the model was calculated. For the Precision-Recall curve, the area under the curve was measured. The higher the area under the curve, the better the model performance in identifying cephalopod species from the beak images.

of *S. esculenta* and *L. uyii* perfectly. The performance of the ANN model can be observed through the precision recall-curve (Fig. 4B).

## Deep features

Both upper and lower beak images were used as inputs into CNN models to extract the deep features. Table 7 shows the average testing accuracy of each classifier with deep features extracted. All CNN models achieved good results with the best result of 91.14% (± 0.09),

**Table 7** Performance of eight classifiers with deep features extracted.

| Model | Testing Accuracy* (AUC) | | | | | |
|---|---|---|---|---|---|---|
| | VGG19 | | InceptionV3 | | ResNet50 | |
| | UB | LB | UB | LB | UB | LB |
| ANN | 88.63(0.95) | **91.14(0.96)** | **87.54(0.94)** | 87.49(0.94) | **86.86(0.93)** | 85.77(0.91) |
| SVM | 81.94(0.11) | 88.57(0.94) | 81.54(0.89) | 84.57(0.91) | 79.03(0.90) | 83.89(0.92) |
| RF | 85.31(0.93) | 89.83(0.95) | 84.29(0.90) | 82.97(0.90) | 85.43(0.93) | 83.31(0.90) |
| kNN | 76.97(0.87) | 85.60(0.93) | 76.00(0.85) | 81.09(0.88) | 76.74(0.86) | 79.31(0.87) |
| DT | 58.63(0.73) | 56.63(0.71) | 49.37(0.66) | 50.63(0.65) | 48.11(0.66) | 53.20(0.69) |
| LR | 89.66(0.96) | 91.71(0.95) | 85.77(0.95) | 85.37(0.92) | 84.86(0.93) | 85.83(0.92) |
| LDA | 83.14(0.88) | 86.34(0.91) | 88.86(0.95) | 89.43(0.95) | 82.11(0.90) | 82.34(0.89) |
| GNB | 82.23(0.85) | 86.97(0.89) | 77.54(0.82) | 77.60(0.83) | 67.60(0.72) | 70.11(0.75) |

**Notes.**
*Average testing accuracy from the five-fold CV results with 10 times runs.
UB, Upper beak; LB, Lower beak; AUC, Average area under the precision–recall curve in one of the runs.
Bolded text indicated the best results for each traditional feature model (RF and LDA models showed overfitting as testing accuracy was much lower than the training accuracy).

87.54% (± 0.10), and 86.86% (± 0.09) for VGG19, InceptionV3 and ResNet50 respectively. Figure 5A shows the confusion matrix from one of the runs of the ANN model for deep features extracted from the VGG19 with lower beak images. The ANN model was shown to classify most of the species perfectly, except for *S. esculenta* and *L. uyii.* The performance of the ANN model can be observed through the precision recall-curve (Fig. 5B).

# DISCUSSION

To the best of our knowledge, this automated identification study is the first of its kind using beak images from cephalopods sampled in the region of Southeast Asia. In general, the easiest way to identify a cephalopod species is based on the appearance of the whole-body. This approach however hinges on one having access to a readily available, comprehensive taxonomic key and working with fresh or well-preserved whole specimens. Development of an identification approach using cephalopod hard parts, especially beaks, is fundamentally important to resolve species identity from digested cephalopod within stomachs of predators. Our study takes this approach further to develop an automated classification of beak images, thus widening the toolkit available for species identification with lesser reliance on manual labour.

It should be clarified that the method developed in this study could also be applied to images of whole cephalopod bodies. However, their soft-bodied nature makes them difficult to manipulate, easily broken during sampling, cleaning and photographing processes, as well as rapidly decay. Based on the samples collected in our study, the tiny, long tentacles of the cephalopod were mostly broken prior to the image acquisition process. This may cause some errors in extracting useful information from whole body images, such as shape features of the cephalopods. Therefore, images of beaks instead of whole bodies were chosen to train each classifier to obtain an automated cephalopod species identification model.

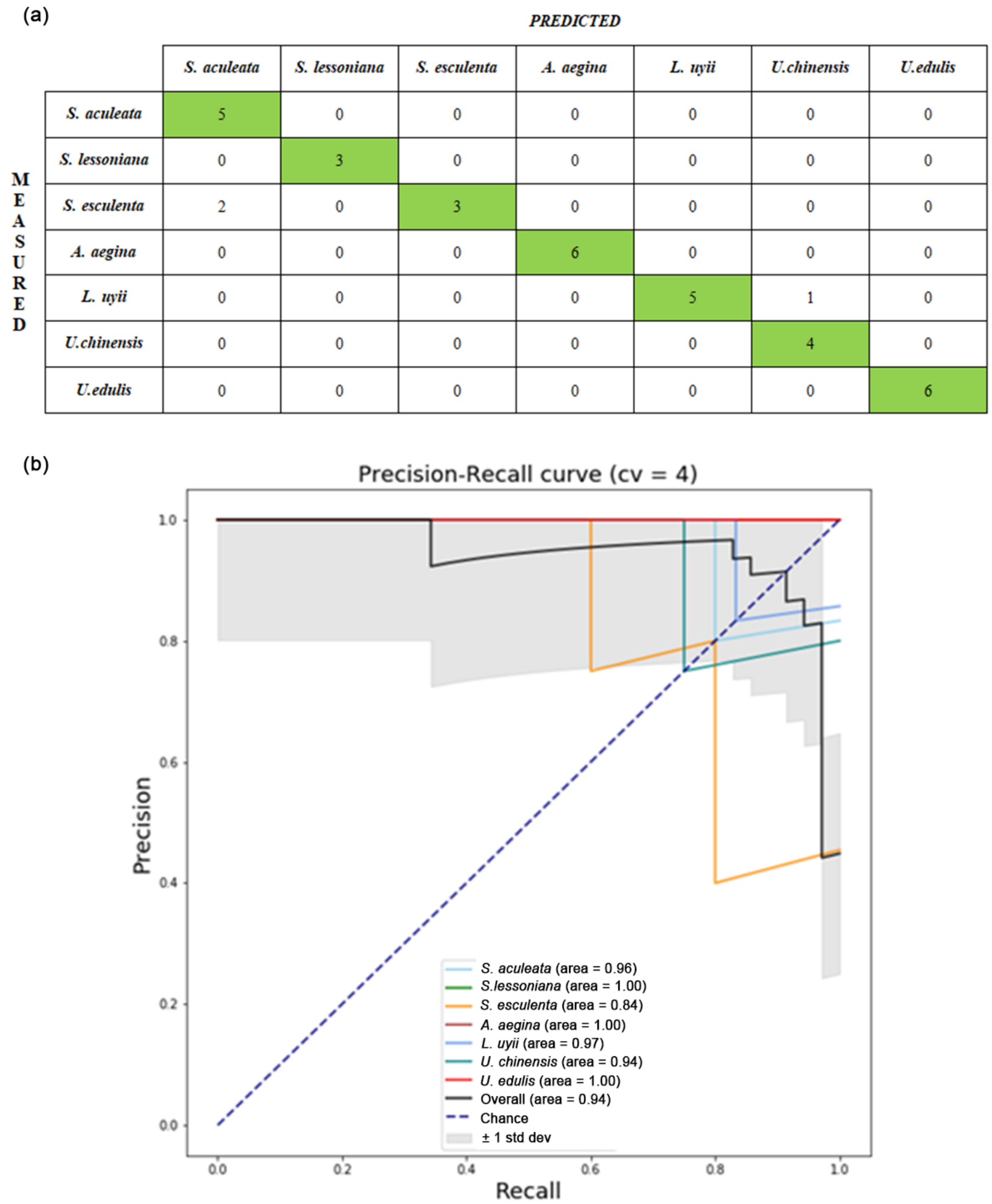
(a)

| | | PREDICTED | | | | | | |
|---|---|---|---|---|---|---|---|---|
| | | S. aculeata | S. lessoniana | S. esculenta | A. aegina | L. uyii | U.chinensis | U.edulis |
| MEASURED | S. aculeata | 5 | 0 | 0 | 0 | 0 | 0 | 0 |
| | S. lessoniana | 0 | 3 | 0 | 0 | 0 | 0 | 0 |
| | S. esculenta | 2 | 0 | 3 | 0 | 0 | 0 | 0 |
| | A. aegina | 0 | 0 | 0 | 6 | 0 | 0 | 0 |
| | L. uyii | 0 | 0 | 0 | 0 | 5 | 1 | 0 |
| | U.chinensis | 0 | 0 | 0 | 0 | 0 | 4 | 0 |
| | U.edulis | 0 | 0 | 0 | 0 | 0 | 0 | 6 |

**Figure 5 Performance evaluation from one of the runs in the VGG19-ANN model of lower beak images: (A) confusion matrix; (B) precision-recall curve.** For the confusion matrix, the precision and recall value of the identification model was computed from the testing set. Each cephalopod species was computed for its precision and recall values to visualize the differences in the performance of the model. The average precision–recall curve of the model was calculated.For the Precision-Recall curve, the area under the curve was measured. The higher the area under the curve, the better the model performance in identifying cephalopod species from the beak images.

We used a smartphone to capture the beak images as this is a readily available tool that can be used by both researchers and citizen scientists. The smartphone was relatively easy to handle and could be focused on the small-sized beak and adjusted for specific angles. During the image acquisition, the left lateral view of the upper beak and lower beak was taken and used in training the cephalopod species identification model. The left lateral view of the beaks was found to provide more distinctive features among the seven cephalopod species collected. The low-quality images did render the image pre-processing more challenging,

specifically the actual edge of the beak and the background could not be differentiated. Some noises were left and may affect the quality of information extracted. Nevertheless, the image segmentation process through the thresholding method had efficiently eliminated the noises left in the images and increased the accuracy of the features extracted from the beak images.

From the results shown in Table 6, the colour feature was found to be better than shape features in differentiating the beaks extracted from each cephalopod species. The HOG descriptor had obtained the best accuracy of 68.69% using ANN and lower beak images. The main reason for this is due to the limited shape features extracted from the beak images which led to difficulty in resolving the tiny differences between the beaks of each cephalopod species. Also, the colour HOG gave better results than the grey HOG descriptor in extracting the colour features from the cephalopod beak images. The gradient or colour changes of the species-specific beaks were significantly distinguished by the three colour channel images (RBG) of the colour HOG descriptor instead of the single colour channel (grey-level intensity) of the grey HOG descriptor. The better performance of the beak colour feature for species identification can also be explained by the close relationship between the beak's darkening stages and age for each cephalopod family (*Clarke, 1962*).

The model performance of the selected classifier increased by combining two out of the three single descriptors. The hybrid descriptor of (colour HOG + MSD) obtained the highest accuracy of 73.09% with the ANN model and lower beak images. The combination of the colour and the shape features had provided more details in differentiating the seven cephalopod species from the beak image provided. The hybrid features can best describe the differences between the beaks since the colour information or the shape information can support each other and result in a higher accuracy of the identification model. The ANN model worked better with hybrid descriptors as the ANN classifier usually performed the best in classification problems which involve small sample sizes (*Pasini, 2015*).

Incorporation of the deep features (Table 7) greatly improved the accuracy of the identification model. The pre-trained CNN model helped to scan the beak images, searched for features that were correlated to the classification problem and combined all the selected features as the input feature vector. Most of the classifiers could achieve accuracy of more than 85% with the deep features, except for DT. The lower accuracy of the DT model was due to the insufficient samples provided. DT model tended to overfit as compared to other classifiers and resulted in lower accuracy (*Liu et al., 2005*). This was because the DT model used only one tree to make the node splitting and generalization. However, by increasing the number of trees, such as in RF, it could overcome the weakness of DT. Since the sample size of this study was small for each cephalopod species, only the important features that can best describe the differences between the cephalopod species were required to train the classifiers. Too many features in the input may cause overfitting, *i.e.,* a well fitted model with data provided but with weak power of generalisation and low accuracy of prediction (*Loughrey & Cunningham, 2004*). For example, the RF and LDA models showed overfitting, as testing accuracy was much lower than the training accuracy. This overfitting problem could be minimized with larger sample sizes for future studies (*Steyerberg, 2019*).

**Table 8 Comparison of previous and current study.**

| Study | Sample | Methodology | | Results |
|---|---|---|---|---|
| | | Feature extraction | Classification | |
| *Orenstain et al. (2016)* | Seven specimens of *Sepia officinalis* (cuttlefish) | Texton-based (mottle and pattern) of cuttlefish | SVM | Best accuracy = 94% |
| *Himabindu, Jyothi & Mamatha (2017)* | 50 squid species | Size measurements of mantle, fin and head | ANN | Best accuracy = 98.6% |
| *Jin et al. (2017)* | Three Loliginidae Squid Species (256 samples) | Extract feature from statolith and beak | Geometric outline with PCA and SDA | Accuracies between 75.0%–88.7% |
| Current study | Seven cephalopod species (174 samples) | Traditional features and deep features of upper and lower beaks | ANN, SVM, RF, kNN, DT, LR, LDA, GNB | Best accuracy = 91.14%. |

The performance of this study was compared to some related previous studies as shown in Table 8. These studies either involved only one type of cephalopod species (*Orenstain et al., 2016*), or using the whole-body morphometric measurements of squids (*Himabindu, Jyothi & Mamatha, 2017*) and there were no deep features extracted (*Jin et al., 2017*; *Himabindu, Jyothi & Mamatha, 2017*; *Orenstain et al., 2016*). The most distinct advantage of this study is the introduction of the deep features in identifying the cephalopod species using upper and lower beaks. From the results of this study, the deep features are better suited in describing the characteristics of cephalopod beaks of each species than the traditional features.

Our study comparing lower and upper beak images is in concordance with findings from other studies that lower beaks are more useful in species identification for cephalopods (*Richoux et al., 2010*; *Xavier, Phillips & Cherel, 2011*). However, for applications of the model to quantify cephalopod prey contributions, improving species identification from upper beaks remain an area of priority due to differing numbers of lower and upper beak samples in studies of the same predator stomachs (*Xavier, Phillips & Cherel, 2011*). Future work in evaluating the performance of the beak images for identification of cephalopods from stomach contents should also focus on evaluating accuracy of identification on both fresh and old samples of beaks due to varying degree of erosion of the beak materials (*Xavier et al., 2015*).

Our limited sampling from the two locations within the Strait of Malacca over a short time duration originally yielded 18 putative species of cephalopods; however only five cuttlefishes, five squids, and three octopus could be identified to the species level out of seven cuttlefishes, six squids, and five octopus putative species respectively (*Muhammad, 2017*). Seven of these cephalopod species had sufficient sample sizes which allowed their inclusion in this machine learning study. The best estimated species richness of cephalopods for the Strait area is 33 species from a global study that focused only on coastal cephalopods (*Rosa et al., 2019*). For context, the Strait is one of the three hotspot ecoregions for cephalopod species richness within the Indo-Pacific area, and inshore squids made up 11 out of the 33 species, and this species richness is likely an underestimation (*Rosa et al., 2019*). Thus the number of species included in the automated identification model developed in this work

represented about 20% of the best species richness estimate. In general, classification error would reduce with increasing sample size per class, or species in this case (*Jain AK & Mao, 2000*). Therefore, future research work should focus on increasing images per species and including the unsampled species within the model development and validation.

## CONCLUSIONS

Traditional features and deep features were extracted from beak images of seven cephalopod species and evaluated using eight classifiers. This study found that ANN achieved the best testing accuracy of 91.14% with deep features extracted by the VGG19 from the lower beak images. Deep features performed better than the traditional features and lesser pre-processing works are needed for deep feature extraction. However, there are some limitations in this proposed model which included unbalanced and limited sample size, a single view of beak included and the limited number of shape features in the MSD descriptors. Hence, future works should include increasing the species variety and the number of samples, adding more shape features such as convex area, eccentricity, and Euler number, and also evaluation of other CNN models. These approaches may help to recognize the minor beak differences between the cephalopod species by increasing details in the extracted features.

## ACKNOWLEDGEMENTS

Sincerely thanks go to Mr. Muhammad Azwan Bin Yahaya from Programme Biodiversity and Ecology, the postgraduate students from the Institute of Advanced Studies, UM and the staff from IOES for help provided during the sample collection.

### Funding

This study was supported by the Universiti Malaya RU Grant (RU009H-2020), the Universiti Malaya Top 100 Universities in The World Fund (TU001-2018), and the Universiti Malaya Research Grant (RP018C-16SUS). The funders had no role in study design, data collection and analysis, decision to publish, or preparation of the manuscript.

### Grant Disclosures

The following grant information was disclosed by the authors:
The Universiti Malaya RU Grant: RU009H-2020.
The Universiti Malaya Top 100 Universities in The World Fund: TU001-2018.
The Universiti Malaya Research Grant: RP018C-16SUS.

### Competing Interests

The authors declare there are no competing interests.

## Author Contributions

- Hui Yuan Tan and Zhi Yun Goh performed the experiments, analyzed the data, prepared figures and/or tables, authored or reviewed drafts of the paper, and approved the final draft.
- Kar-Hoe Loh conceived and designed the experiments, analyzed the data, authored or reviewed drafts of the paper, and approved the final draft.
- Amy Yee-Hui Then and Hasmahzaiti Omar analyzed the data, authored or reviewed drafts of the paper, and approved the final draft.
- Siow-Wee Chang conceived and designed the experiments, performed the experiments, analyzed the data, prepared figures and/or tables, authored or reviewed drafts of the paper, and approved the final draft.

## Data Availability

The lower and upper beak images from the seven cephalopod species used in this study are available in the Supplemental Files.

## Supplemental Information

Supplemental information for this article can be found online at http://dx.doi.org/10.7717/peerj.11825#supplemental-information.

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
