# Peer review of "Cephalopod species identification using integrated analysis of machine learning and deep learning approaches"

_PeerJ, doi:10.7717/peerj.11825_

## Round 0.1 · original submission · Major Revisions

Dear authors,

Two external reviewers have accessed your manuscript "Cephalopod species identification using integrated analysis of machine learning and deep learning" providing the comments that are reported herein. As you can see, they both found your study worth of publication but have suggested major and minor revision. They identified a number of minor issues that would require careful revision before this paper is recommended for acceptance. I agree with them and suggest a language revision before re-submission.

Best,

Wagner Magalhães
PeerJ Academic Editor

·

Basic reporting

Thank you for opportunity of reviewing the paper entitled “Cephalopod species identification using integrated analysis of machine learning and deep learning”.

The manuscript deals with machine learning techniques applied to taxonomic identification of cephalopods from Malaysian waters, including octopus, squid and cuttlefish species. Following rigorous tests of both upper and lower beak images as possible tools for identification, results show that the use of latter yielded identification with ~ 91% of confidence.

The manuscript has merit to be published, but some limitations must be observed prior final acceptance. First, the manuscript would definitively benefit from a professional English language service. Second, the use of lower beaks as reliable tool for cephalopod identification is not new and these structures have been used at least since the 1960s (Clarke, 1962, 1980), so authors must acknowledge this in the Introduction and Discussion.

The discussion was very methodology-orientated, and failed in addressing the usefulness of the reported findings in the wider taxonomic context.

Experimental design

No comment

Validity of the findings

No comment

Additional comments

You methods and results are clear and well-described. Thus, I focused my review on the Discussion.

Your discussion was very methodology-orientated, and I was really expecting you tell to the readers the importance of your findings in improving the use of lower beaks images for species identification. The use of lower beaks for cephalopod identification from stomach contents is not new and these structures are traditionally used for this purpose (please see Clarke, 1962, 1980 just for start – there are plenty of similar studies out there). Unless I missing something, I could not find any mention of these early (and recent) studies along your whole manuscript.

There were other important aspects you failed to mention. For instance, how your method performs in identifying correctly morphologically-similar species of the same family and genera as Uroteuthis chinensis and U. edulis? For the sake of exemplification: if one has a sample of 100 beaks of these two species mixed together in a stomach content sample, how good is your method in correctly identifying each species in that sample?

You should definitively expand the discussion regarding the performance comparison between your method and the results achieved by other authors (lines 395-397). It is neither informative nor useful (for the sake of discussion) just “calling” a comparative table (Table 7, in your case) and make a brief (and poor) comparison. This is not enough. You have sufficient and decent data to describe what is better/innovative in your results in relation to the published results from other authors. You also should recognize the limitaions of your method [for instance, the ANN method employed by Himabindu et al. (2017) had a higher accuracy than your results].

References

Clarke M (1962a) Significance of cephalopod beaks. Nature, 193,560-1
Clarke MR (1986) A handbook for the identification of cephalopod beaks. Clarendon Press, Oxford
Himabindu K, Jyothi S, Mamatha D (2017) Classification of squids using morphometric measurements. Gazi University Journal of Science, 30(2), 61-71.

Reviewer 2 ·

Basic reporting

The manuscript aim to compare eight machine learning methods for image classification of cephalopods. They argue that the comparison is relevant since this has not been done for cephalopods in Malaysia and that cephalopods are economically important for fisheries in the region as well as being an important prey in the marine food web.

Since the aim of the study is to determine which classification method is best, it would be relevant to include paragraphs in the introduction and discussion that address the literature that make similar comparisons of methods for species classification or more general image classifications. I.e. not limiting the comparison to classification of squid species as you did in Table 7.

Experimental design

No comment

Validity of the findings

Although it may be difficult to increase sample size, the limitations of a small dataset should be discussed. This might impact the both test results and the reliability of applications of the model to classify new material. I would suggest to perform some kind of sensitivity analysis that show the robustness.

For a biological material I also find it surprising that the dataset is not more unbalanced. I.e. more species with very few examples and a few number of species with most examples. Did you make additional steps to balance the dataset?

It is not clear from the manuscript if the number of species included are close to the number of known species in the region? You should also address how a relatively low number of species (classes) in the model affect the results.

---

## Round 0.2 · Major Revisions

Dear Dr. Chang,

Many thanks for re-submitting your manuscript. I apologize for the delay in getting this review back to you. Your manuscript went back to an additional reviewer who has suggested more changes. I hope you can accommodate these additional edits that include additional explanation in the methods section, improvement of the figures and table's captions, and most importantly, a thorough review of the English language.

I agree with the reviewer that a professional Academic English review should be done prior to a re-submission of this manuscript.

[·

Basic reporting

Thank you for the opportunity of (re-) reviewing this manuscript. The present version certainly improved on the early draft analysed, but I found an important taxonomic flaw, and I also made suggestions to improve the text presentation and clarity. These are described in the "General comments for the author" section below.

That said, I would strongly encourage the authors to seek a professional Academic English review service to improve their text. In my opinion, the English of this manuscript is not yet good enougth for publication in an international Journal such as PeerJ.

Experimental design

No comments here.

Validity of the findings

Their findings are relevant and certainly would improve cephalopod taxonomic resolution in poorly surveyed regions or in counries that lack taxonomic experts on the group.

Additional comments

ABSTRACT

Line 43:

Please, change “Besides, deep features were extracted (…)” to “In addition, deep features were extracted (…)”

INTRODUCTION

Lines 61–62:

Please, change “(…) which includes octopus, squid and cuttlefish.” to “(…) which includes octopus, squid, cuttlefish and nautiluses.”

MATERIALS & METHODS

You must briefly describe here the steps showed in the flowchart depicted in Figure 1. Something like: “This flowchart is described as follows: “Sample collection” show that squid, octopus and cuttlefish samples were acquired from local fisheries at two landing sites off western Malaysia. “Image Acquisition” depicts how images were acquired (…)”... And so on.

Lines 176–177:

Please make sure to use the appropriate “times” (×) symbol.

Thus, change “5312px x 2988px” to “5312px × 2988px” and “532px x 299px” to “532px × 299px”

Make sure to double-check this detail for the rest of your text.

Lines 189 and 190:

Please use the “end dash” (–) symbol to show value intervals.

Thus, change “(0 - 239)” to “(0–239)” and “(240 - 255)” to “(240–255)”.

Make sure to double-check this detail for the rest of your text.

Line 216:

Please, change “Besides (…)” to “In addition to (…)”

Line 246:

The “n” must be in italics here.

Make sure to double-check this detail for the rest of your text.

Line 278:

Please change “. 5-fold cross-validation (CV) (…)” to “Five-fold cross-validation (CV) (…)” (A number must no begin a sentence!).

Make sure to double-check this detail for the rest of your text.

RESULTS

Line 303:

Sepioteuthis lessoniana is NOT a cuttlefish but a true squid species belonging to the same family (Loliginidae) of the three squids you recognized as such (i.e. Loliolus uyii, Uroteuthis chinensis and U. edulis). That’s a flaw that certainly affected the interpretation of your results! You MUST reanalise your results under the ligth of this observation and modify whatever you find necessary accordingly.

Line 330:

The taxonomic nomenclature rules forbid abbreviate a species’ name. Thus, you must change “S. esc” to “S. esculenta” here and in your Figures 4 and 5. You must also write the complete species name of each one of the remaining cephalopod species listed in Figures 4 and 5.

Make sure to double-check this detail for the rest of your text.

TABLES

Table 1: You must include a brief description of each one of these ten features to the reader. For example, what exactly “Solidity / convexity” means?

Table 5: you must explain in the table caption what does means the values marked in bold.

Any table or figure in a scientific text must be self-informative! Please, improve your table captions!

FIGURES

You must explain in the figure caption how to interpret the precision recall-curve. I would also recommend you explain it briefly, when you call Figures 4 and 5 in your Results. Remember that not all readers are familiar with your statistical techniques/methods. Thus, these details must be explained for the sake of clarity.

In addition, please double-check the captions of all remaining figures. As I said before, any table or figure in a scientific text must be self-informative.

---

## Round 0.3 · Minor Revisions

Dear authors,

Please see the attached PDF for additional edits and suggestions for the manuscript.

---

## Round 0.4 · accepted · Accept

Dear authors,

Many thanks for addressing my suggestions to your manuscript.